# Examination of an averaging method for estimating repulsion and attraction interactions in moving groups

**Rajnesh K. Mudaliar[1,2], Timothy M. Schaerf** [1] *

**1** School of Science and Technology, University of New England, Armidale, NSW, Australia, **2** School of Mathematical and Computing Science, Fiji National University, Suva, Fiji

* tschaerf@une.edu.au

**Data Availability Statement:** All computer codes required to generate the data for this study (via simulation), and then analyse this data, have been provided with the submission.

## Abstract

Groups of animals coordinate remarkable, coherent, movement patterns during periods of collective motion. Such movement patterns include the toroidal mills seen in fish shoals, highly aligned parallel motion like that of flocks of migrating birds, and the swarming of insects. Since the 1970's a wide range of collective motion models have been studied that prescribe rules of interaction between individuals, and that are capable of generating emergent patterns that are visually similar to those seen in real animal group. This does not necessarily mean that real animals apply exactly the same interactions as those prescribed in models. In more recent work, researchers have sought to infer the rules of interaction of real animals directly from tracking data, by using a number of techniques, including averaging methods. In one of the simplest formulations, the averaging methods determine the mean changes in the components of the velocity of an individual over time as a function of the relative coordinates of group mates. The averaging methods can also be used to estimate other closely related quantities including the mean relative direction of motion of group mates as a function of their relative coordinates. Since these methods for extracting interaction rules and related quantities from trajectory data are relatively new, the accuracy of these methods has had limited inspection. In this paper, we examine the ability of an averaging method to reveal prescribed rules of interaction from data generated by two individual based models for collective motion. Our work suggests that an averaging method can capture the qualitative features of underlying interactions from trajectory data alone, including repulsion and attraction effects evident in changes in speed and direction of motion, and the presence of a blind zone. However, our work also illustrates that the output from a simple averaging method can be affected by emergent group level patterns of movement, and the sizes of the regions over which repulsion and attraction effects are apparent can be distorted depending on how individuals combine interactions with multiple group mates.

**Funding:** This work was supported by funding from the Australian Research Council under project DP190100660. There was no additional external funding received for this study.

**Competing interests:** The authors have declared that no competing interests exist.

# 1 Introduction

There are many perceived advantages for animals to stay and move in groups, including: reduced individual probability of becoming a victim of predation in the event of an attack on the group (via a dilution effect [1] and a confusion effect [2, 3]); the enhanced ability of many eyes to monitor the environment for potential threats at group level [4]; and enhanced decision-making ability during foraging [5], and finding and establishing a new home [6, 7]. Movements of animal groups occur for many reasons, including seasonal migration, abundance of forage and movement from one home site to another [8]. Animal groups undergoing collective motion often form striking group level patterns of movement, including the complex swirls of starling murmurations, toroidal milling patterns produced by shoals of fish, the chaotic but nevertheless guided motion of honey bee swarms, and directed parallel motion seen in groups of many species [9, 10].

It has been hypothesised that the group level patterns of collective motion emerge due to simple interactions between group members [11]. This fundamental insight into the possible nature of the mechanisms that drive the patterns of collective motion, and interest in these patterns across multiple scientific disciplines, including biology, physics, mathematics and computer science, has led to the development of a number of models of such motion [12–26]. Many of these models prescribe how individuals adjust their velocity based on the relative positions and velocities of their group mates according to some combination of the following broad "rules of interaction":

1. Repulsion: individuals adjust their velocity to avoid collision with near neighbours.

2. Orientation: individuals adjust their velocity to match that of neighbours that are nearby (but not close enough to crash into).

3. Attraction: to avoid group fragmentation, individuals adjust their velocity to move towards other group members that are somewhat removed from the individual's current relative position in the group.

With such broad rules in action, collective motion models are capable of generating emergent patterns that are visually similar to those seen in real animal groups, including coordinated parallel motion, milling and swarming [13, 22].

The success of collective motion models in generating realistic looking motion has led to these models being the dominant method for understanding collective movement. However, the fact that prescribed within-model interactions of the sort listed above generate realistic global motions does not necessarily mean that real animals apply such interactions. A sequence of papers starting in 2008 has sought to infer the nature of rules of interaction in moving animal groups directly from trajectory data, starting with a large study of natural flocks of starlings [27, 28], and then moving to other species, such as surf scoters, [29], and fish [30, 31]. Here the trajectory data is a time series of coordinates in two or three dimensions for each individual, gathered by either automated visual tracking methods (examples include Ctrax [32] and idTracker [33]) applied to video or sequences of still images, or via Global Positioning System (GPS) technology (as was used by Nagy et al. [34]).

A variety of methods have now been employed to extract interaction rules, including averaging methods (sometimes referred to as force-matching methods) [30, 31, 35, 36], analyses of burst and coast dynamics [37, 38], and function fitting via machine learning algorithms [39]. In one of the simplest formulations, averaging methods determine the mean changes in the direction of motion and speed of an individual as a function of the relative coordinates of group mates. While these methods for extracting interaction rules and related

quantities from trajectory data are relatively new and becoming popular, the accuracy of these methods needs more scrutiny. An exception is in the recent work of Heras et al. [39], where a deep attention network is used to fit functions that describe the rules of interaction of zebra fish. Heras et al. [39] validate their artificial neural network methods for reconstructing rules of collective motion using simulated data from models similar to those of [13, 16], and variants on these models where individuals interact over topological, rather than metric, length scales. In more recent work, [38] discussed a number of potential short-comings of the averaging (force-matching) approach, but did not interrogate the method using simulated data.

In this paper, we examine how well an averaging method captures rules of interaction from trajectory data generated via simulations with prescribed model based rules. Here, we seek to use data of similar duration, and representative of similar numbers of individuals, to that derived from experimental studies where averaging methods have been used as part of the analysis. The approach used to examine the averaging method here is quite general, and could have been applied using data generated by any individual based model for collective motion where the positions of individuals are tracked explicitly, irrespective of the species modelled. The approach could also be modified to validate the use of an averaging method to infer interactions in three dimensions, using data generated by any number of models that operate in three dimensions (see for example [13, 23, 24, 40]). For this work, however, we performed simulations in two spatial dimensions using two well established models for collective motion: the zonal self-propelled individual simulation model developed in [13], and the ordinary differential equation (ODE) model studied by [21, 22, 26]. We first describe the averaging methods used in this study, which are derived from those described in [31, 36], in Section 2. These methods fit functions that describe the average response of an individual in terms of changes in speed and direction of motion over time as a function of the relative coordinates of group mates, and the speed of the individual. In Section 3 we outline our simulations, along with associated parameter values and the emergent group level behaviours that we observed. As part of our analysis, we examine the effects of the number of individuals, the overall duration of simulations, and the use of only the first or last half of data sets on the accuracy of the averaging method. Results and a discussion follow in sections 4 and 5 respectively. The two individual based models used for this study are detailed in S2 Section of the S1 File for this paper. It is possible to obtain some fundamental analytic results for these models that give an explicit form for pairwise interactions in terms of changes in speed and direction of motion. We illustrate these results in S3 Section of the S1 File, to be used as a point of comparison to the interactions inferred from simulated data, however, we note that *a priori* we do not expect exact extraction of the pairwise interactions by the averaging methods that we use due to the way data is aggregated across multiple individuals.

## 2 Averaging method for estimating interaction rules from trajectory data

We applied the methods described in [36] to determine the mean change in angle of motion over time, and the mean change in speed over time, of individuals as a function of the coordinates of group mates relative to both the location and direction of motion of each individual. In addition, for analysis of data generated by the ODE model in particular, we examined both mean changes in speed and direction of motion of individuals as a function of the relative coordinates of group mates and the speed of the focal individual. As the focus of this study is the accuracy of the methods described in [36] we detail the necessary calculations in the following subsections.

## 2.1 Fundamental measures of movement

We estimated the components of the velocities of each individual from simulated data using the standard forward-difference approximations:

$$u_i(t) = \frac{x_i(t + \Delta t) - x_i(t)}{\Delta t} \qquad \text{and} \qquad v_i(t) = \frac{y_i(t + \Delta t) - y_i(t)}{\Delta t} \qquad (2.1)$$

where $(x_i(t), y_i(t))$ is the position of individual $i$ at time $t$, and $\Delta t$ is the duration between consecutive simulation time steps. The components of the unit vector in the direction of an individual's velocity, $\mathbf{V}_i = u_i(t)\mathbf{i} + v_i(t)\mathbf{j}$, for each individual $i$, for each time $t$, are defined as

$$\hat{u}_i(t) = \frac{u_i(t)}{||\mathbf{V}_i(t)||} \qquad \text{and} \qquad \hat{v}_i(t) = \frac{v_i(t)}{||\mathbf{V}_i(t)||} \qquad (2.2)$$

where the norm $||\mathbf{V}_i(t)||$ is defined as

$$||\mathbf{V}_i(t)|| = \sqrt{u_i(t)^2 + v_i(t)^2}. \qquad (2.3)$$

We used the components of velocity to find the magnitude of the change in direction of motion of each individual $i$ from time $t$ to time $t + \Delta t$ via:

$$\psi_i(t) = \cos^{-1}(\hat{u}_i(t)\hat{u}_i(t + \Delta t) + \hat{v}_i(t)\hat{v}_i(t + \Delta t)). \qquad (2.4)$$

We also determined the sense of rotation of the individual explicitly, that is whether the individual turned clockwise or anticlockwise at each time step. To do this we examined the sign of the vertical component of the cross product of the unit velocity vectors of each individual $i$ at times $t$ and $t + \Delta t$. Individual $i$ turned anticlockwise (clockwise) as it moved from time $t$ to time $t + \Delta t$ if the sign of the following equation is positive (negative):

$$\lambda_i(t) = \text{sgn}\,(\hat{u}_i(t)\hat{v}_i(t + \Delta t) - \hat{u}_i(t + \Delta t)\hat{v}_i(t)). \qquad (2.5)$$

where sgn is the sign function.

Taking into account whether the individual $i$ turns clockwise or anticlockwise, the signed change in direction of motion over time in degrees is given by:

$$\frac{\Delta\theta_i}{\Delta t}(t) = \frac{180}{\pi}\begin{cases} \lambda_i(t)\psi_i(t)/\Delta t & \text{if } \lambda_i(t) \neq 0, \\[2mm] \psi_i(t)/\Delta t & \text{if } \lambda_i(t) = 0. \end{cases} \qquad (2.6)$$

We estimated the speed of individual $i$ at time $t$ directly from the components of velocity via:

$$s_i(t) = \sqrt{u_i(t)^2 + v_i(t)^2}. \qquad (2.7)$$

We then approximated the change in speed over time of individual $i$ using

$$\frac{\Delta s_i}{\Delta t}(t) = \frac{s_i(t + \Delta t) - s_i(t)}{\Delta t}. \qquad (2.8)$$

## 2.2 Relative coordinates of group mates

We then examined the average change in speed over time, $\Delta s/\Delta t$, and the average change in direction of motion over time, $\Delta\theta/\Delta t$, as a function of the relative $(x, y)$ coordinates of group mates alone, as well as the $(x, y)$ coordinates of group mates and the speed of the focal

individual. First we detemined the distance between a focal individual *i* (every individual was treated as a focal individual in turn) and every other individual *j* in the group, for all times *t*, using the distance formula:

$$d_{i,j}(t) = \sqrt{(x_j(t) - x_i(t))^2 + (y_j(t) - y_i(t))^2}. \tag{2.9}$$

Then we calculated the angle between the unit velocity vector of the focal individual *i* and the straight line segment from individual *i* to individual *j*. The unit vector pointing along the line segment from individual *i* to individual *j* has components:

$$\hat{x}_{ij}(t) = \frac{x_j(t) - x_i(t)}{d_{i,j}(t)} \qquad \text{and} \qquad \hat{y}_{ij}(t) = \frac{y_j(t) - y_i(t)}{d_{i,j}(t)}. \tag{2.10}$$

The unsigned angle between the direction of motion of individual *i* and the unit vector pointing from individual *i* to individual *j* is:

$$\phi_{ij}(t) = \cos^{-1}\left(\hat{u}_i(t)\hat{x}_{ij}(t) + \hat{v}_i(t)\hat{y}_{ij}(t)\right). \tag{2.11}$$

We then employed a similar technique to that summarised by Eq (2.5) to determine if individual *j* lay to the left or right of individual *i*. Relative to the direction of motion of individual *i*, individual *j* lies to the left (right) of individual *i* if the sign of the following equation is positive (negative):

$$\zeta_{i,j}(t) = \text{sgn}\left(\hat{u}_i(t)\hat{y}_{ij}(t) - \hat{v}_i(t)\hat{x}_{ij}(t)\right). \tag{2.12}$$

Taking into account whether individual *j* is on the left or right of individual *i* and combining Eqs (2.11) and (2.12) we find the signed angle between the direction of motion of individual *i* and the unit position vector of individual *j* relative to focal individual *i*:

$$\vartheta_{i,j}(t) = \begin{cases} \zeta_{i,j}(t)\phi_{i,j}(t) & \text{if } \zeta_{i,j}(t) \neq 0, \\ \phi_{i,j}(t) & \text{if } \zeta_{i,j}(t) = 0. \end{cases} \tag{2.13}$$

Hence, Eqs (2.9) and (2.13) give the polar coordinates of the position of an individual *j* relative to individual *i*'s position and direction of motion, $(d_{i,j}(t), \vartheta_{i,j}(t))$. We then converted these polar coordinates to rectangular coordinates using:

$$x_{ij,relative}(t) = d_{i,j}(t)\cos(\vartheta_{i,j}(t)),$$
$$y_{ij,relative}(t) = d_{i,j}(t)\sin(\vartheta_{i,j}(t)). \tag{2.14}$$

In the above $(x_{ij,relative}(t), y_{ij,relative}(t))$ relative coordinate system, the focal individual *i* is located at the origin, $(0, 0)$, with its velocity vector aligned with the positive *x*-axis. We divided a square local domain centred on the focal individual into a set of overlapping square bin regions (details in S1.1 Section of the S1 File). For each focal individual *i*, partner *j*, and discrete time *t*, we then stored the changes in speed, $\frac{\Delta s_i}{\Delta t}(t)$, and direction of motion, $\frac{\Delta \theta_i}{\Delta t}(t)$, in all bins that contain $(x_{ij,relative}(t), y_{ij,relative}(t))$, or where necessary for our calculations, in bins that also took into account the speed of the focal individual (S1.2 Section of the S1 File). Once all the data for a measure of interest was binned according to the appropriate independent variables ($(x, y)$ or $(x, y, s)$), we then determined the mean value in each bin. Once this process was complete, we rendered the resulting fitted function using MATLAB's *surf* function.

## 3 Numerical investigation

### 3.1 Overview of models

We applied the models developed by [13] and [22] to simulate the trajectories of individuals involved in collective movement in two dimensions. The ultimate goal of this simulation work was to examine the ability of the method detailed in section 2 to reconstruct the model prescribed interactions directly from the trajectories. Both models, and all methods of analysis, were encoded in MATLAB.

Previous related work on the analysis of midge swarms (*Cladotanytarsus atridorsum*) included steps to correct for effects of group level patterns of movement in examining correlations between the midges [41]. The emergent states of both the models that we used can be dependent on initial conditions, with multiple stable group-level patterns of movement possible for the same set of within-model parameters, as well as there being a dependence on parameters of which states are possible, [13]. We therefore decided to examine how emergent state might affect the results inferred by the method of Section 2.

We applied the self-propelled particle zonal model of [13] in one if its simplest forms, where $N$ simulated individuals travelled at constant speed, $s$, and modified their direction of motion to avoid collisions with neighbours at close range (within their Zone of Repulsion (ZOR), with radius $r_r$), oriented their velocity with neighbours at intermediate distances (within their Zone of Orientation (ZOO), with outer radius $r_o$ and width $\Delta r_o$), and moved towards neighbours at greater distances (within their Zone of Attraction (ZOA), with outer radius $r_a$ and width $\Delta r_a$) (see Fig 1). The model acts on a discrete time scale with equally spaced time steps, $\Delta t$, admits a blind region behind the individuals of angular extent $\omega_{blind}$, constrains the turning ability of individuals via a maximum turning rate, $\theta$, and allows for some random variation in the chosen directions of individuals, controlled via the parameter $\eta$. Simulations were performed over an unbounded domain in two-dimensions. Full mathematical details of the model are provided in S2 Section of the S1 File.

The ODE model developed in [22] has built-in mechanisms for increases and decreases in individual speed. The model traces the movement of $N$ individuals whose changes in position and velocity are described by:

$$
\begin{cases}
\dfrac{d\mathbf{x}_i}{dt} = \mathbf{v}_i, & (i = 1, \ldots, N) \\[2mm]
\dfrac{d\mathbf{v}_i}{dt} = (\alpha - \beta|\mathbf{v}_i|^2)\mathbf{v}_i - \dfrac{1}{N}\sum_{j \neq i}\nabla U(|\mathbf{x}_i - \mathbf{x}_j|), & (i = 1, \ldots, N).
\end{cases}
\tag{3.1}
$$

where $\mathbf{x}_i$ and $\mathbf{v}_i$ are the position and velocity of the $i$-th individual respectively. $\alpha$ and $\beta$ are non-negative parameters, $\alpha$ models the self-propulsion of individual $i$ and $\beta$ is the friction parameter for individual $i$. $U$ in the above system of ODEs is the *Morse potential* [22], defined by

$$
U(|\mathbf{x}_i - \mathbf{x}_j|) = -C_A e^{\frac{-|\mathbf{x}_i - \mathbf{x}_j|}{l_A}} + C_R e^{\frac{-|\mathbf{x}_i - \mathbf{x}_j|}{l_R}},
\tag{3.2}
$$

where $|\mathbf{x}_i - \mathbf{x}_j| = \sqrt{(x_j - x_i)^2 + (y_j - y_i)^2}$. $C_A$ and $C_R$ are the amplitudes of the attraction and repulsion effects, and $l_A$ and $l_R$ relate to the ranges of attraction and repulsion respectively.

It is possible to derive some basic analytical results that describe pairwise interactions in both the models used in this study in a form that is directly comparable to outputs from the averaging methods described in Section 2. The results include formulae that describe the change in direction of motion of a focal individual as a function of the relative coordinates of a

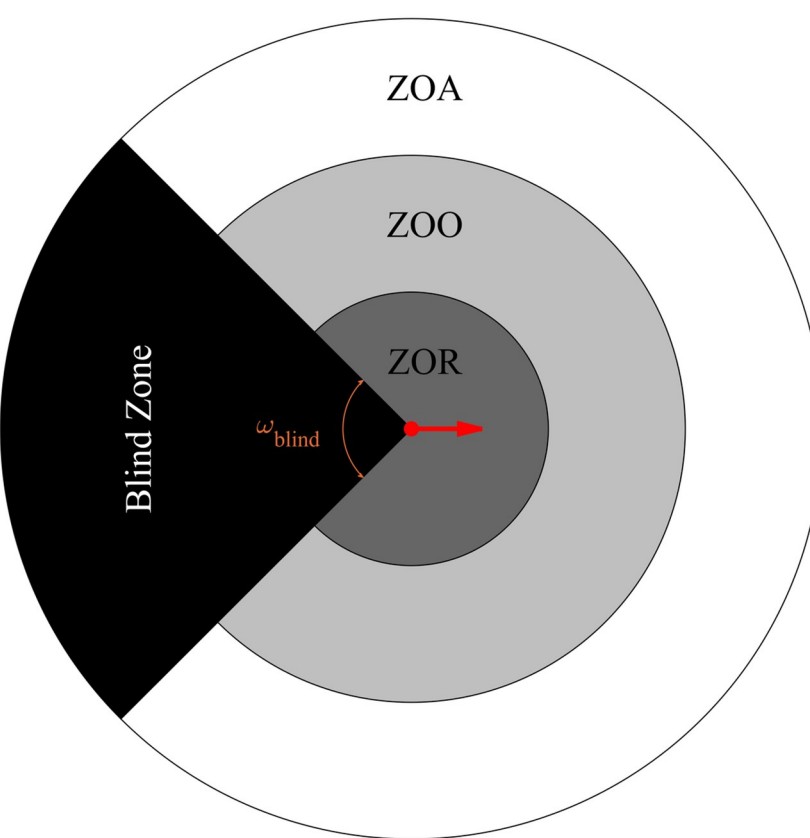

**Fig 1. In the zonal model developed in [13], a focal individual (located at the center of the diagram, moving to the right in the direction indicated by the red arrow) is assumed to adjust its direction of motion to: Move away from group members in the dark gray zone (the Zone of Repulsion (ZOR)) to avoid collision, align its direction of motion with those in the light gray zone (the Zone of Orientation (ZOO)), and move towards individuals within the white circle (the Zone of Attraction (ZOA)) to remain in contact with the group.** The individual will not adjust its motion in response to neighbours located in its blind zone (indicated by the black wedge). $\omega_{blind}$ is the blind angle. The radius of the circle bounding the ZOR is $r_r$, the radius of the circle bounding the ZOO is $r_o$ and the radius of the circle bounding the ZOA is $r_a$. These circles are concentric.

single group mate for the self-propelled particle model (ignoring effects of noise), the change in speed of an individual as a function for the relative coordinates of a single group mate for the ODE model, and the change in direction of motion of an individual as a function of its own speed and the relative coordinates of it group mate, again for the ODE model. The analytical pairwise interactions for both models are detailed in S3 Section of the S1 File, and are used as a point of comparison against the analysis of the simulated data using the averaging method.

## 3.2 Simulations and analysis of data from zonal model

We performed simulations using the model of [13] for varying sizes of repulsion, orientation, and attraction zones, and differing implementations of individual blind zones. The bulk of our simulations focussed on groups of $N = 25$ individuals, but we also examined potential group size effects on our trajectory analysis over a more limited set of simulations with $N = 10$ and $N = 40$ individuals. We also varied the duration over which simulations were performed, first performing short duration simulations of 1000 time steps, and then followed up these calculations with equivalent simulations of 10000 time steps in duration. The latter, 10000 time step

**Table 1. Summary of parameters used in zonal model simulations.**

| Parameter | Unit | Symbol | Values Used |
|---|---|---|---|
| **Couzin Model Parameters** | | | |
| Number of individuals | None | $N$ | 10, 25, 40 |
| Zone of repulsion | units | $r_r$ | 0.5, 1, 1.5, 2 |
| Zone of orientation | units | $\Delta r_o$ | 0.01, 1, 1.5, 2, 4.5, 5, 5.5 |
| Zone of attraction | units | $\Delta r_a$ | 8, 11, 12.99 |
| Blind angle | Degrees | $\omega_{blind}$ | 0, 90 |
| Maximum turning rate | Degrees per second | $\theta$ | 40 |
| Individual speed | Units per second | $s$ | 3 |
| Time step increment | Seconds | $\Delta t$ | 0.1 |
| The standard deviation in noise | rads | $\eta$ | 0.1 |

duration simulations have a similar length (in terms of time steps) to the data used in some empirical studies, such as [36], which had 9000 time steps of data per experimental trial for groups of 8 individuals. We also note that in the original study of [13], most simulations had reached a stable emergent pattern of movement within 5000 time steps (albeit for groups of 100 individuals). The simulations were performed for all possible combinations of zone sizes shown in Table 1 below, such that $r_r < r_o < r_a$ and $r_r + \Delta r_o + \Delta r_a = 14$ for each differing implementation of blind zone. Ten simulations were performed for each combination of zone sizes and associated emergent states; we chose this number of replicates because it is comparable to the number of experimental replicates performed per treatment in experimental studies that have used the averaging method [36]. Table 1 also lists other important parameter values used in our simulations.

Table 2 shows the different zone sizes used and the emergent collective behaviour produced by the individuals with differing forms of blind zone for the short duration simulations of 1000 time steps. The emergent states were classified by visual inspection for the short duration simulations. We labelled emergent collective states that included swarming and milling behaviour as states or patterns exhibiting cohesion. S1 Table in the S1 File lists the data subsets and their emergent collective behaviour for simulations performed for 10000 time steps. We performed a more algorithmic classification of emergent states for the 10000 time step calculations based on order parameters that measured and summarised the instantaneous agreement in direction of motion of individuals (the polarisation, $p_{group}$), and the agreement in the sense of rotation of group members about the group centre (the angular momentum, $m_{group}$), [13, 42], over the second half of each simulation. In addition we analysed the fragmentation of groups using the algorithm described in [43, 44]. (See S4 Section of the S1 File for further details on the calculation of the order parameters and the analysis of group fragmentation). We followed the broad classification scheme adopted in [42], such that groups (that did not fragment) with $p_{group} > 0.65$ and $m_{group} < 0.35$ were classified as exhibiting parallel aligned movement, and swarming ($p_{group} < 0.35$, $m_{group} < 0.35$) and milling ($p_{group} < 0.35$, $m_{group} > 0.65$) groups were classified together as exhibiting cohesion (without parallel movement).

For all simulations, we saved the position data $(x_i(t), y_i(t))$ for each individual at each time step. We then analysed this trajectory data using the methods described in Section 2, in particular to examine changes in direction of motion as a function of relative partner coordinates for each of the data subsets in Table 2 and S1 Table in S1 File. We only conducted analysis of fragmenting groups when these represented the dominant or only form of emergent behaviour for

**Table 2. Summary of zone size, size of blind region and respective emergent collective behaviour for simulations with 1000 time steps.**

| $r_r$ | $\Delta r_o$ | $\Delta r_a$ | Form of blind zone as given | Emergent Pattern |
|---|---|---|---|---|
| 0.5 | 0.51 | 12.99 | In ZOO, ZOA and ZOR $\omega_{blind} = 90°$ | cohesion |
| 0.5 | 2.5 | 11 | In ZOO, ZOA and ZOR $\omega_{blind} = 90°$ | cohesion |
| 0.5 | 5.5 | 8 | In ZOO, ZOA and ZOR $\omega_{blind} = 90°$ | cohesion/ parallel aligned |
| 1 | 0.01 | 12.99 | In ZOO, ZOA and ZOR $\omega_{blind} = 90°$ | cohesion |
| 1 | 2 | 11 | In ZOO, ZOA and ZOR $\omega_{blind} = 90°$ | cohesion |
| 1 | 5 | 8 | In ZOO, ZOA and ZOR $\omega_{blind} = 90°$ | cohesion/ parallel aligned |
| 1.5 | 1.5 | 11 | In ZOO, ZOA and ZOR $\omega_{blind} = 90°$ | cohesion |
| 1.5 | 4.5 | 8 | In ZOO, ZOA and ZOR $\omega_{blind} = 90°$ | cohesion/ parallel aligned |
| 2 | 1 | 11 | In ZOO, ZOA and ZOR $\omega_{blind} = 90°$ | cohesion |
| 0.5 | 0.51 | 12.99 | In ZOO, ZOA and ZOR $\omega_{blind} = 0°$ | cohesion |
| 0.5 | 2.5 | 11 | In ZOO, ZOA and ZOR $\omega_{blind} = 0°$ | cohesion/ parallel aligned |
| 0.5 | 5.5 | 8 | In ZOO, ZOA and ZOR $\omega_{blind} = 0°$ | parallel aligned |
| 1 | 0.01 | 12.99 | In ZOO, ZOA and ZOR $\omega_{blind} = 0°$ | cohesion |
| 1 | 2 | 11 | In ZOO, ZOA and ZOR $\omega_{blind} = 0°$ | cohesion |
| 1 | 5 | 8 | In ZOO, ZOA and ZOR $\omega_{blind} = 0°$ | parallel aligned |
| 1.5 | 1.5 | 11 | In ZOO, ZOA and ZOR $\omega_{blind} = 0°$ | cohesion |
| 1.5 | 4.5 | 8 | In ZOO, ZOA and ZOR $\omega_{blind} = 0°$ | cohesion/ parallel aligned |
| 2 | 1 | 11 | In ZOO, ZOA and ZOR $\omega_{blind} = 0°$ | cohesion |
| 0.5 | 0.51 | 12.99 | In ZOO, ZOA $\omega_{blind} = 90°$, in ZOR $\omega_{blind} = 0°$ | cohesion |
| 0.5 | 2.5 | 11 | In ZOO, ZOA $\omega_{blind} = 90°$, in ZOR $\omega_{blind} = 0°$ | cohesion |
| 0.5 | 5.5 | 8 | In ZOO, ZOA $\omega_{blind} = 90°$, in ZOR $\omega_{blind} = 0°$ | cohesion/ parallel aligned |
| 1 | 0.01 | 12.99 | In ZOO, ZOA $\omega_{blind} = 90°$, in ZOR $\omega_{blind} = 0°$ | cohesion |
| 1 | 2 | 11 | In ZOO, ZOA $\omega_{blind} = 90°$, in ZOR $\omega_{blind} = 0°$ | cohesion |
| 1 | 5 | 8 | In ZOO, ZOA $\omega_{blind} = 90°$, in ZOR $\omega_{blind} = 0°$ | cohesion/ parallel aligned |
| 1.5 | 1.5 | 11 | In ZOO, ZOA $\omega_{blind} = 90°$, in ZOR $\omega_{blind} = 0°$ | cohesion |
| 1.5 | 4.5 | 8 | In ZOO, ZOA $\omega_{blind} = 90°$, in ZOR $\omega_{blind} = 0°$ | cohesion/ parallel aligned |
| 2 | 1 | 11 | In ZOO, ZOA $\omega_{blind} = 90°$, in ZOR $\omega_{blind} = 0°$ | cohesion |

a given set of parameter values (as listed in S1 Table of the S1 File). In addition, for the 10000 time step simulations, we repeated the analysis, but applied to only the first or second half of each set of data to examine the potential effects of transitions from initial to emergent states.

## 3.3 Estimating the radius of the ZOR directly from graphical output

We fitted a circle through three distinct points on what appears to be the region of greatest repulsion interactions and determined its radius using the method described in [36] for fitting circles. The region of largest apparent repulsion effects might be identifiable from the graphs of the mean change in direction of motion over time as the area closest to the focal individual (at (0, 0)) covered by both a blue region on the left of the focal individual and a red region on the right of the focal individual. The apparant blind zone manifests in the graphs of the mean change in direction of motion over time as an approximately triangular shape behind the focal individual in the region of strongest turning mediated repulsion effects. We estimated the extent of such apparent blind angles using the graph-based approach detailed in S9.1 Section of the S1 File. Fig 2 illustrates the points used to estimate the circle bounding an apparent region of repulsion, along with points used to estimate the extent of the blind angle trailing an individual for a specific case.

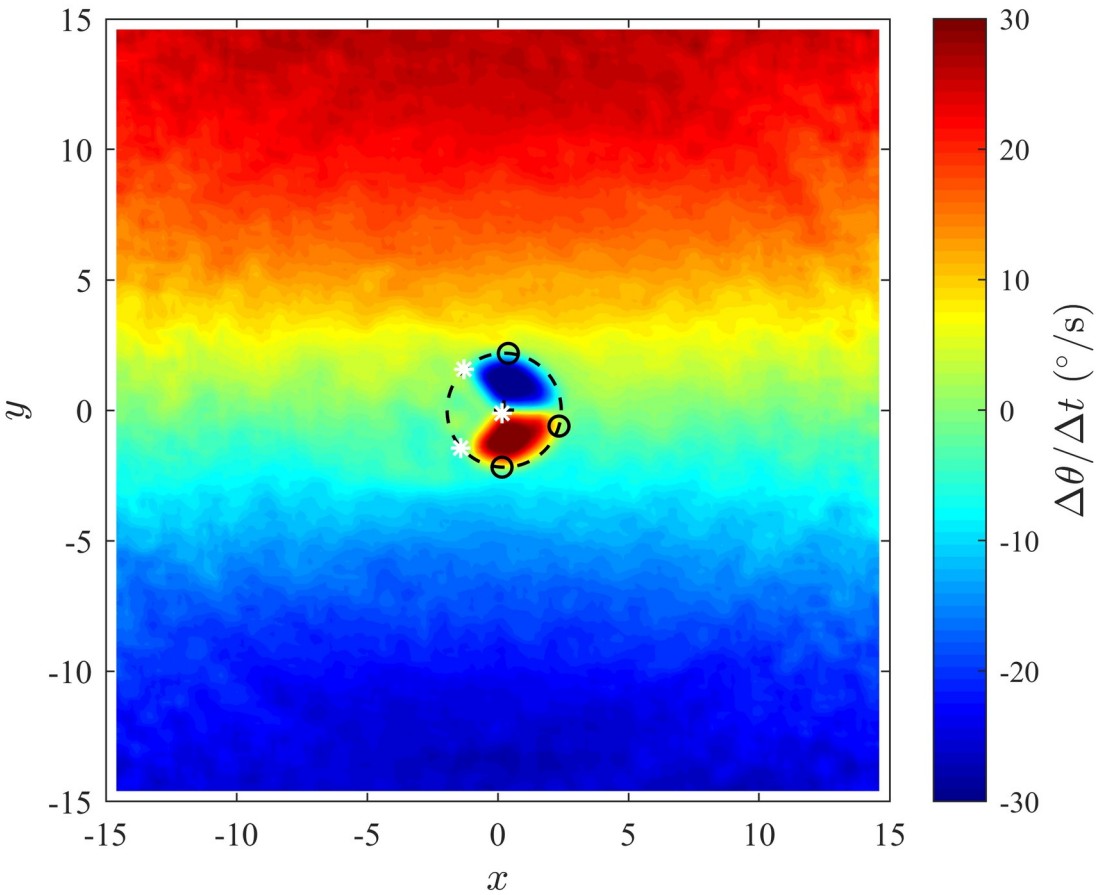

**Fig 2. Changes in direction as a function of the relative coordinates of partners in groups with emergent cohesion behaviour where $\omega_{blind} = 90°$, $r_r = 2$, $\Delta r_o = 1$ and $\Delta r_a = 11$.** Small black circle represents the points used to fit the circle estimating the ZOR. The white asterisks represent the points that were used to estimate the size of the blind angle. (Derived from simulations with $N = 25$ individuals over 1000 time steps).

### 3.4 Simulations and analysis of data from the ODE model

We determined numerical solutions to the system 3.1 using a standard fourth-order Runge-Kutta scheme for time integration. As with the zonal model of [13], we obtained data from a number of simulations for different combinations of within-model parameters. Table 3 shows

**Table 3. Modelling parameters and emergent collective motion patterns for the ODE model with $N = 10$ individuals.**

| item | $\alpha$ | $\beta$ | $C_A$ | $C_R$ | $l_A$ | $l_R$ | number of simulations | Emergent Behaviour |
|------|----------|---------|-------|-------|-------|-------|------------------------|---------------------|
| a | 0.15 | 0.05 | 100 | 50 | 100 | 20 | 10 | double mill |
| b | 0.04 | 0.005 | 100 | 150 | 100 | 3 | 80, 80, 40 | anticlockwise mill, clockwise mill, swarm |
| c | 1 | 1 | 100 | 50 | 50 | 5 | 80, 40 | parallel aligned, swarm |
| d | 1 | 0.5 | 100 | 50 | 200 | 30 | 80 | swarm |

For item (a) a double mill is an annular structure where group members simultaneously traversed the annulus in clockwise and anticlockwise directions. For item (b), emergent states were dependent on initial conditions, and it was possible to generate an anticlockwise mill, clockwise mill and swarm. For item (b) 80 simulations were performed for each sense of milling pattern and 40 simulations were performed for swarms. For item (c), initial condition dependent emergent states were parallel aligned motion and swarm-like behaviour; we performed simulations until we had data for 80 parallel aligned groups, and 40 swarm-like groups.

the different within-model parameters that were used in the simulations and the associated emergent behaviour. The emergent states were identified via visual inspection. Simulations were performed with ten individuals ($N = 10$) for 10000 time steps with time step $\Delta t = 0.1$ for the bulk of our calculations, however we examined group size effects for both parallel aligned and swarming groups with the parameters listed under item (c) of Table 3, including calculations for $N = 5$, $N = 15$ and $N = 25$ individuals. Individuals were initially uniformly randomly distributed within a square region of side length 100 units, with a uniformly distributed random direction of motion on $(0, 2\pi)$, and initial speed set to $\sqrt{\alpha/\beta}$, for all simulations. As was the case with the zonal model, all simulations were performed over an unbounded domain. Previous work with the ODE model suggests that emergent behaviour stabilises by around 2000 simulation time steps, [45], albeit for larger groups with $N = 200$, and for calculations run with time steps of 0.05 units. The same study also notes that the model of [22] typically behaves independent of initial conditions, [45], but for our work we found multiple group-level patterns of movement emerging for the same set of model parameters (as noted in Table 3). Where possible we sought to generate data from at least 10 realisations (and up to 80) representative of a particular emergent state for given parameter values (Table 3).

We used the methods described in Section 2 to examine changes in direction over time of an individual as a function of relative partner positions at different speed intervals, and changes in speed over time of an individual as a function of partner positions for different speed intervals, for each of the emergent states under each of the items listed in Table 3.

## 4 Results

### 4.1 Changes in direction of motion in the zonal model

Results of our analysis of the zonal model, as listed in Table 2 and S1 Table in S1 File are shown in Figs 3 and 4, and S6 to S21 Figs in the S1 File. In addition, Fig 5 illustrates the effects of analysing longer time series (the effects of limiting analysis to the first or second half of each set of trajectories appears in S23 to S25 Figs in the S1 File), and Fig 6 illustrates group size effects for short duration simulations (with longer duration simulations represented by S22 Fig in S1 File). In all these plots the focal individual is located at the origin, moving to the right parallel to the x-axis. Across our analysis, at short range the blue region close to the left of the focal individual indicates that the focal individual turns clockwise, away from partners in this region. Similarly, the red region close to the right of the focal individual indicates that the focal individual turns anticlockwise, away from the partners located closer to the right. The tendency of the focal individual to turn away from closer partners is consistent with the repulsion rule prescribed by the zonal model (equation (S2.1), S2 Section in S1 File). The circle formed by enclosing the red region close to the right and the blue region close to the left of the focal individual in the Figs 3 and 4 (and S6 to S21 Figs in S1 File) is a visual estimate of the ZOR, as identified by the averaging method described in Section 2.

The red region further to the left of the focal individual indicates that the focal individual turns anticlockwise, towards partners in this region. The blue region further to the right of the focal individual indicates that the focal individual turns clockwise, towards partners within that region. The overall tendency to turn towards partners outside the apparent zone of repulsion, which is closer to the focal individual, is consistent with turning mediated attraction to partners in the ZOA, as prescribed in the zonal model [13].

When applying the averaging method to multiple individuals, the exact sizes of the ZOR and the ZOA do not appear to be captured; this is most apparent when making direct comparisons between prescribed pairwise interactions (panel A in each of Figs 3 and 4, and S6 to S21 Figs in S1 File), and the output from the averaging method. The averaging method does

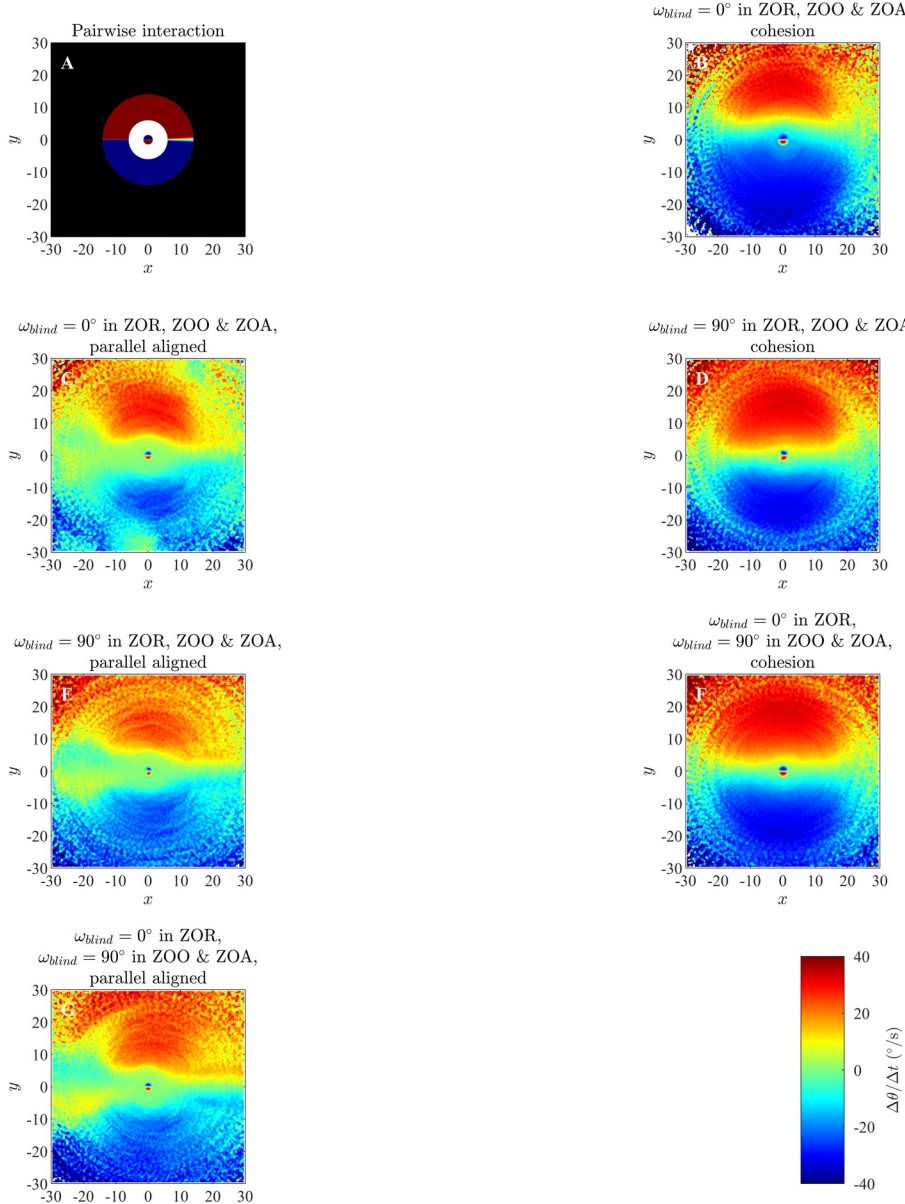

**Fig 3.** Panel A: analytical pairwise interactions for given parameter values, as described in S3.1 Section, where turning of the individuals is governed by equations (S3.5) and (S3.10). Panels B, C, D, E, F and G illustrate changes in direction of motion of individuals as a function of the relative positions of partners obtained via analysis of simulations with $r_r$ = 1.5, $\Delta r_o$ = 4.5 and $\Delta r_a$ = 8 using the averaging method. Positive changes in angle of motion indicate a turn to the left by the focal individual, whereas negative changes in angle of motion indicate a turn to the right. (Derived from simulations with $N$ = 25 individuals over 1000 time steps).

however capture differences in detail across simulations where there was no blind zone, or implementation of the blind zone differed (applying across all zones, or only throughout the ZOO and ZOA), most evidently in a wedge-like region immediately behind the focal individual that is present when there is a blind-zone across all three interaction zones. The wedge-like region is absent when there is no blind zone (contrast plots within Fig 4 to see this in clearest detail).

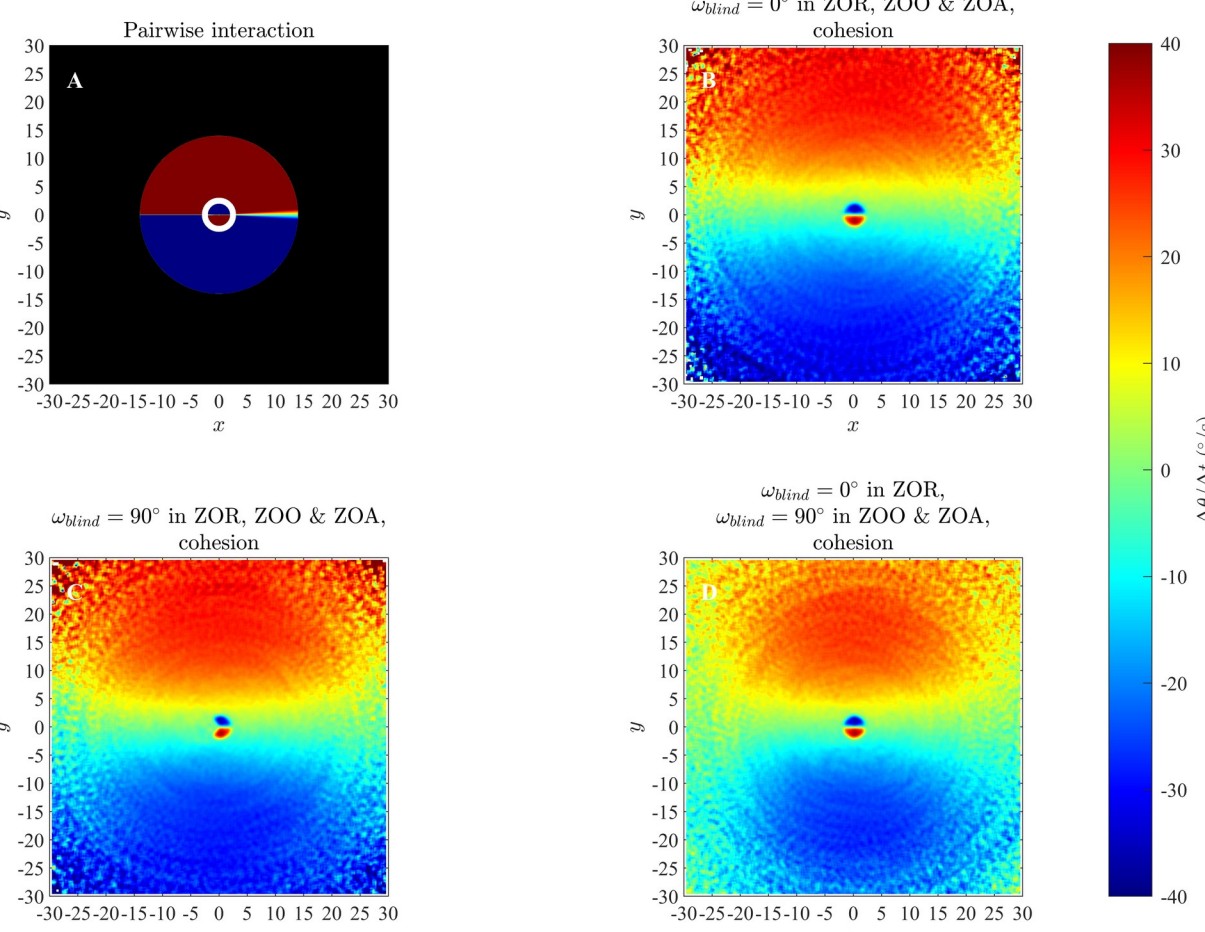

**Fig 4.** Panel A: analytical pairwise interactions for given parameter values, as described in S3.1 Section, where turning of the individuals is governed by equations (S3.5) and (S3.10). Panels B, C and D illustrate changes in direction of motion of individuals as a function of the relative positions of partners obtained via analysis of simulations with $r_r = 2$, $\Delta r_o = 1$ and $\Delta r_a = 11$ using the averaging method. (Derived from simulations with $N = 25$ individuals over 1000 time steps).

In the figures associated with the parallel aligned motion of individuals there is an approximately annular green patch in between the regions of repulsion- and attraction-like behaviour; an equivalent annular region is less apparent (or non-existent) for the groups exhibiting cohesion, but not parallel motion. The size of the annular region seems to correlate with the prescribed size of the ZOO, but the methods used in this paper don't explicitly seek or examine alignment interactions of the sort prescribed in the zonal model [13]. In addition, for simulations run over the longer 10000 time step duration in particular, individuals tended to enact repulsion-like turns away from partners within their blind-zone, but at distances that extended between the outer radius of the zone of orientation and beyond (see S14, S15, S18 and S20 Figs in S1 File). It is not entirely clear what causes this apparent behaviour, but what could be happening is that the turns away from partners are due to the presence of near neighbours in individuals' repulsion zones, with the aggregation of data across all group mates also associating these turns with neighbours that are much further away, and not in the repulsion zone.

In qualitative terms, apart from some additional features in plots generated from the analysis of data derived from 10000 time step simulations, especially the additional apparent

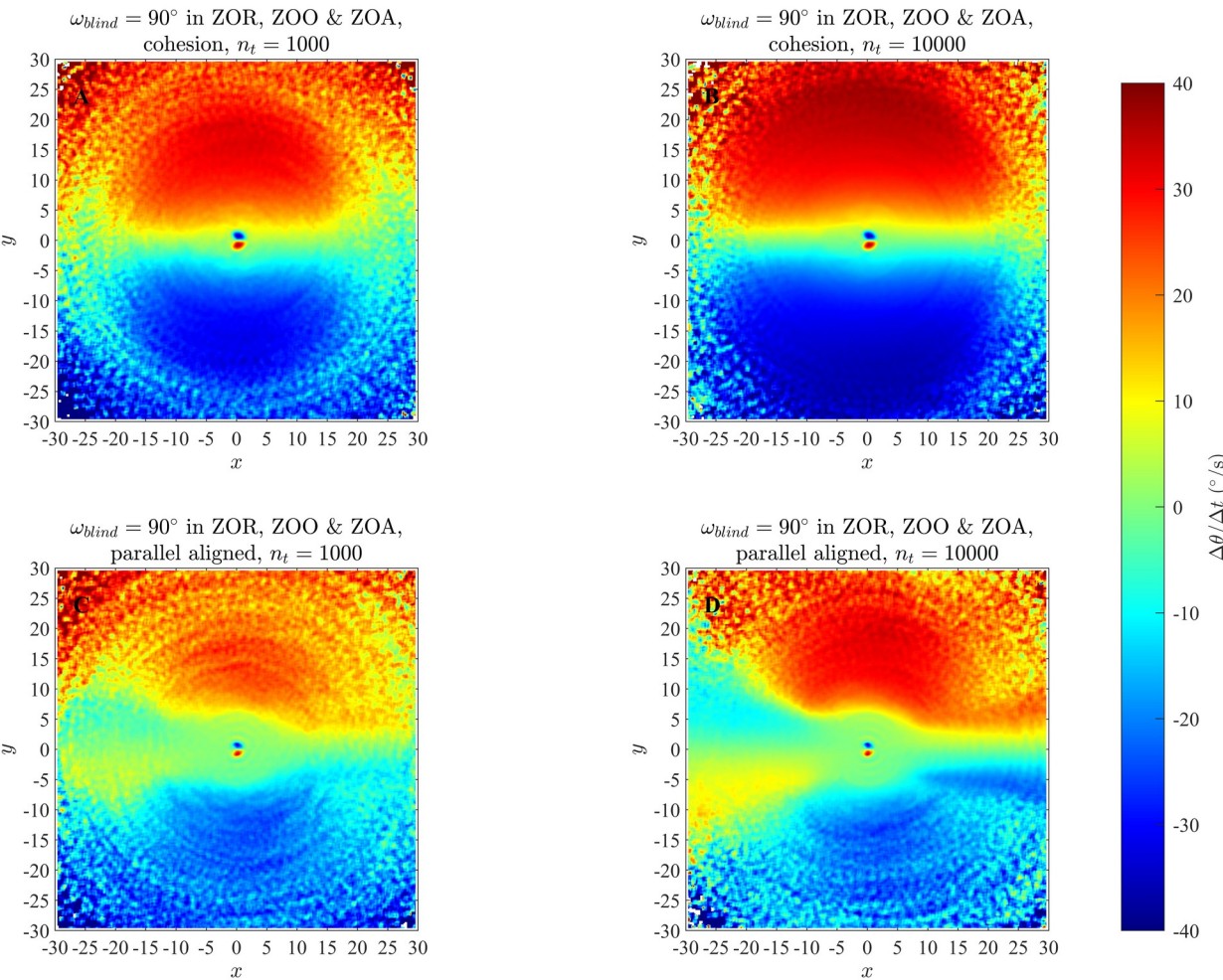

**Fig 5. A comparison of analyses of simulations run for 1000 time steps (left column) and 10000 time steps (right column) for groups that exhibited cohesion without parallel motion (top row) or formed into aligned groups (bottom row) when $r_r$ = 1.5, $\Delta r_o$ = 4.5, $\Delta r_a$ = 8.** (Derived from simulations with $N$ = 25 individuals).

repulsive effects to trailing neighbours in parallel groups mentioned above, the results of analyses of 1000 and 10000 time step simulations are largely similar graphs (Fig 5). Based on the work in [13] and our own experience, emergent behaviour tends to stabilise for most simulations of the zonal model by about 5000 time steps. Hence the first 5000 time steps of our longer duration simulations will include a combination of transient behaviour from the initial distribution of individuals to an emergent state, and perhaps also a susbtantial amount of data in the emergent state. The latter 5000 time steps should represent the behaviour in the emergent state. The qualitative features of the interactions suggested by analysis of the first or last 5000 time steps of the longer duration simulations are very similar, revealing similarly sized regions over which repulsion-like interactions apply at short range, and attraction-like turning behaviour at greater distances from individuals (S23 to S25 Figs in S1 File). A clear difference from the analyses of the first and second halves of the data is that many bins remain unfilled once the group enters, and stays in, a potentially stable pattern of collective movement (S23 to S25 Figs in S1 File, right columns).

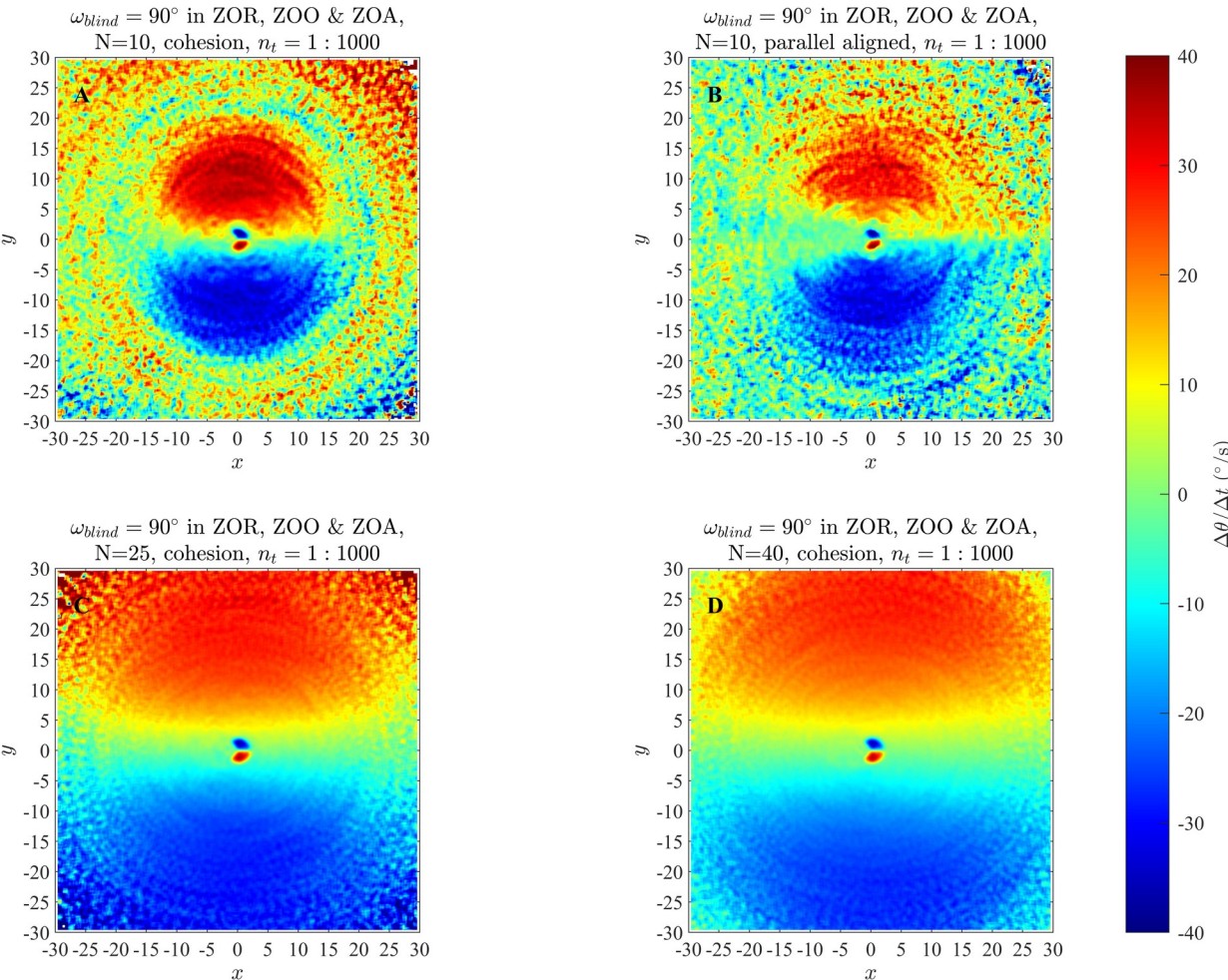

**Fig 6. Changes in direction of motion of individuals as a function of the relative positions of partners obtained via analysis of simulations with $r_r$ = 2, $\Delta r_o$ = 1, $\Delta r_a$ = 11 for groups of $N$ = 10 (panels A and B), $N$ = 25 (panel C), or $N$ = 40 (D) individuals.** Simulations were run for short durations of 1000 time steps.

Increasing group size did not substantially affect the apparent size of the repulsion zone detected by the averaging method (Fig 6 and S22 Fig in S1 File), but the region over which attraction-like behaviour was evident seemed to increase as the number of individuals also increased (Fig 6 and S22 Fig in S1 File).

## 4.2 Estimation of the sizes of blind angles and repulsion zones by visual inspection

Fig 2 & S26 to S31 Figs in S1 File contain the graphs of turning speed as a function of the relative coordinates of other individuals that were used to estimate the size of the blind angle and the radius of the ZOR. The estimated values of the radius of the ZOR and the blind angle are presented in Table 4. The graphs in S26, S27A & S27B, and S28 Figs in S1 File were plotted over a smaller domain and with finer bins compared to the plots in Fig 2, S28B to S31 Figs in S1 File to help resolve the ZOR in our plots when the prescribed radius of the ZOR was 0.5. The estimated value of the radius of the ZOR and the size of the blind angle obtained from the

**Table 4. Summary of estimated blind angles and radii of possible zones of repulsion for respective simulated data with different zone widths for the ZOR, ZOO and ZOA and differing emergent behaviours.**

| | | | | Results for estimating blind angle and radius of zone of repulsion | | |
|---|---|---|---|---|---|---|
| Item | $r_r$ | $\Delta r_o$ | $\Delta r_a$ | Emergent Pattern | Estimated Blind Angle | Estimated Radius of ZOR |
| a | 1 | 0.01 | 12.99 | cohesion | 83.27 | 1.13 |
| b | 1 | 2 | 11 | cohesion | 81.87 | 1.19 |
| c | 1 | 5 | 8 | parallel alligned | 77.74 | 0.79 |
| d | 1 | 5 | 8 | cohesion | 86.82 | 1.21 |
| e | 1.5 | 1.5 | 11 | cohesion | 89.29 | 1.73 |
| f | 1.5 | 4.5 | 8 | cohesion | 88.03 | 1.74 |
| g | 1.5 | 4.5 | 8 | parallel aligned | 88.38 | 1.32 |
| h | 2 | 1 | 11 | cohesion | 89.64 | 2.18 |

The individuals all had a blind angle of $\omega_{blind} = 90°$.

S26, S27A & S27B, and S28A Figs in S1 File, when the radius of the ZOR was $r_r = 0.5$ have been omitted from Table 4 because turning based repulsion effects are not clearly (or approximately) constrained to a circular region.

Table 4 shows estimates of the blind angle and the radius of the ZOR from our graphs. The blind angles that we inferred from our graphs, via computer aided inspection, reasonably approximated the prescribed model blind angle of 90 degrees trailing each individual (Table 4). In addition, our graphically estimated radii for the zone of repulsion were reasonable approximations to the prescribed model radii (Table 4 and Fig 7).

## 4.3 Changes in direction of motion in the ODE model

Results of our analysis for changes in direction of individuals as a function of relative positions of partners for the ODE model, as listed in Table 3, are shown in Figs 8 and 9 (and S32 to S36 Figs in S1 File). In these plots individuals turn away from partners that are in close range and turn towards partners that are at greater distances, which is similar to the repulsion and attraction behaviour of individuals in the zonal model as elaborated in 4.1 Section, and consistent with model prescribed pairwise turning interactions. However, the sizes of the regions over which repulsion-like turning interactions were deduced from simulated data were consistently smaller than those derived from exact pairwise interactions (compare the left and right columns of Figs 8 and 9 (and S32 to S36 in S1 File)), and decreased as group size increased (S42 to S46 Figs in S1 File). In addition, our simulations for the smallest groups (with $N = 5$), did not provide enough data to fill bins in the region closest to the focal individual (white central region in the right column of S42 Fig in S1 File); such a lack of data representing repulsion interactions may be consistent with a drawback of the averaging approach suggested by [38].

Based on examination of Figs 8 and 9, and S32 to S36 Figs in S1 File, it appears that emergent group level patterns of movement have an effect on the turning behaviour inferred by the averaging method. For groups that form anticlockwise rotating mills, the averaging method suggests that individuals tend to turn anticlockwise in response to the relative positions of neighbours located just outside the apparent repulsion zone, irrespective of if these neighbours are positioned to the left or the right of the focal individual (redder regions in the second column of S33 Fig in S1 File). The tendency for anticlockwise turns by the individuals is a reflection of the overall anticlockwise rotation of the group, but does not accurately represent the form of pairwise interactions immediately outside the repulsion zone. Analogous results can

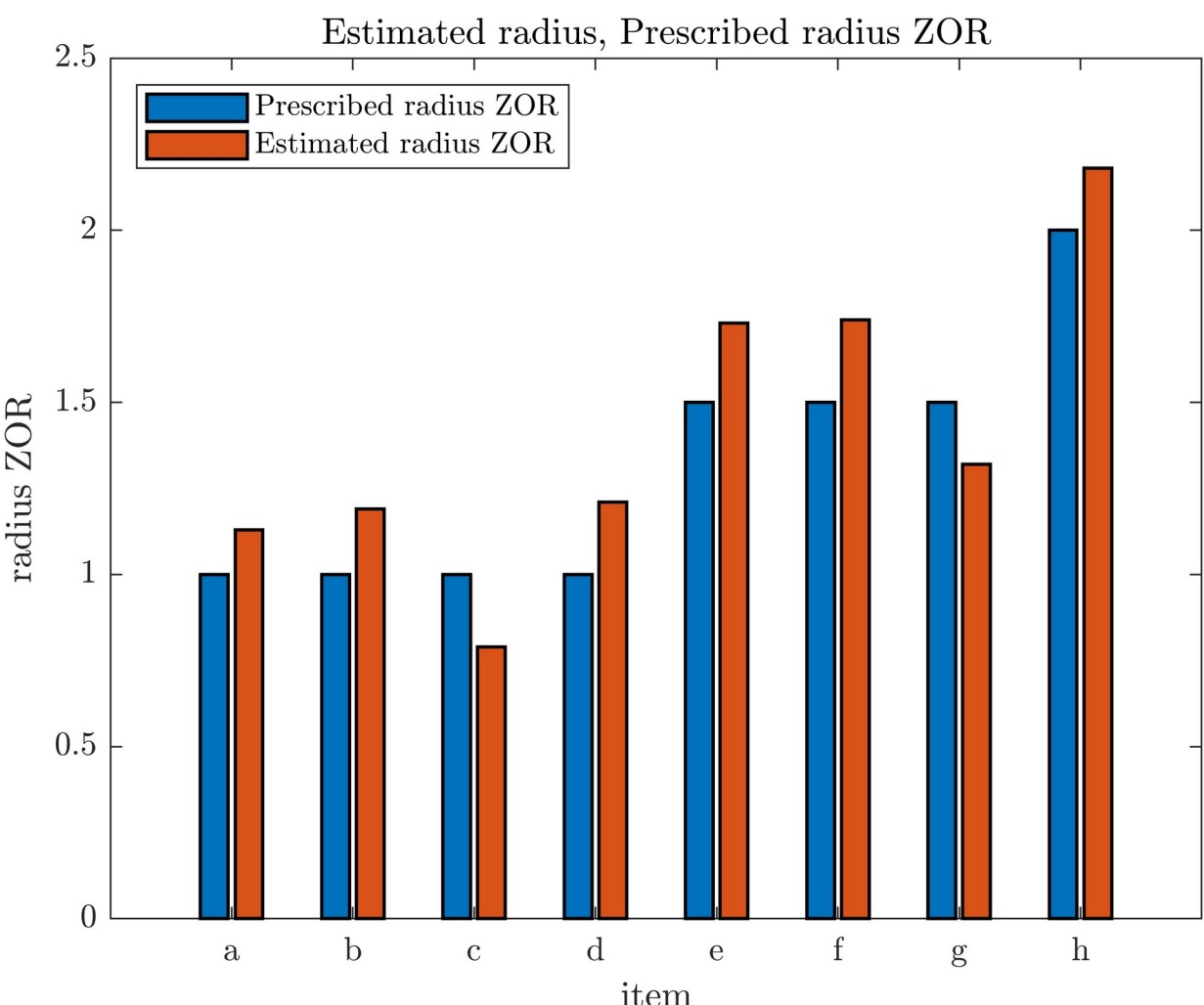

**Fig 7. Comparsion of prescribed radii of the ZOR in simulations and the estimated radii of the same zones from the plots in Fig 2 & S28B-S31 Figs in S1 File.** (Derived from simulations with $N$ = 25 individuals over 1000 time steps).

be seen for groups that formed clockwise rotating mills, with the averaging method suggesting that individuals tended to rotate clockwise in response to near neighbours outside the repulsion zone irrespective of if these neighbours were to the left or right of the focal individual (blue regions just outside the repulsion zones in the right column of S34 Fig in S1 File). For groups that exhibited parallel aligned motion the averaging method suggested that when neighbours occupied a relatively large annular region outside the repulsion zone, then the focal individual did not tend to turn very much in response to the positions of these neighbours (green annular regions outside repulsion zones, right column, Fig 9). We observed similar behaviour in analysis of the zonal model when groups moved in parallel (for example see Fig 3, S7, S8 and S11 Figs), and noted the size of the annular region seemed to correlate with the size of the prescribed orientation zone. However, given that there are no explicit orientation interactions in the ODE model, it seems that the annular region may be indicative of the emergent group structure, rather than the underlying interactions that give rise to this structure.

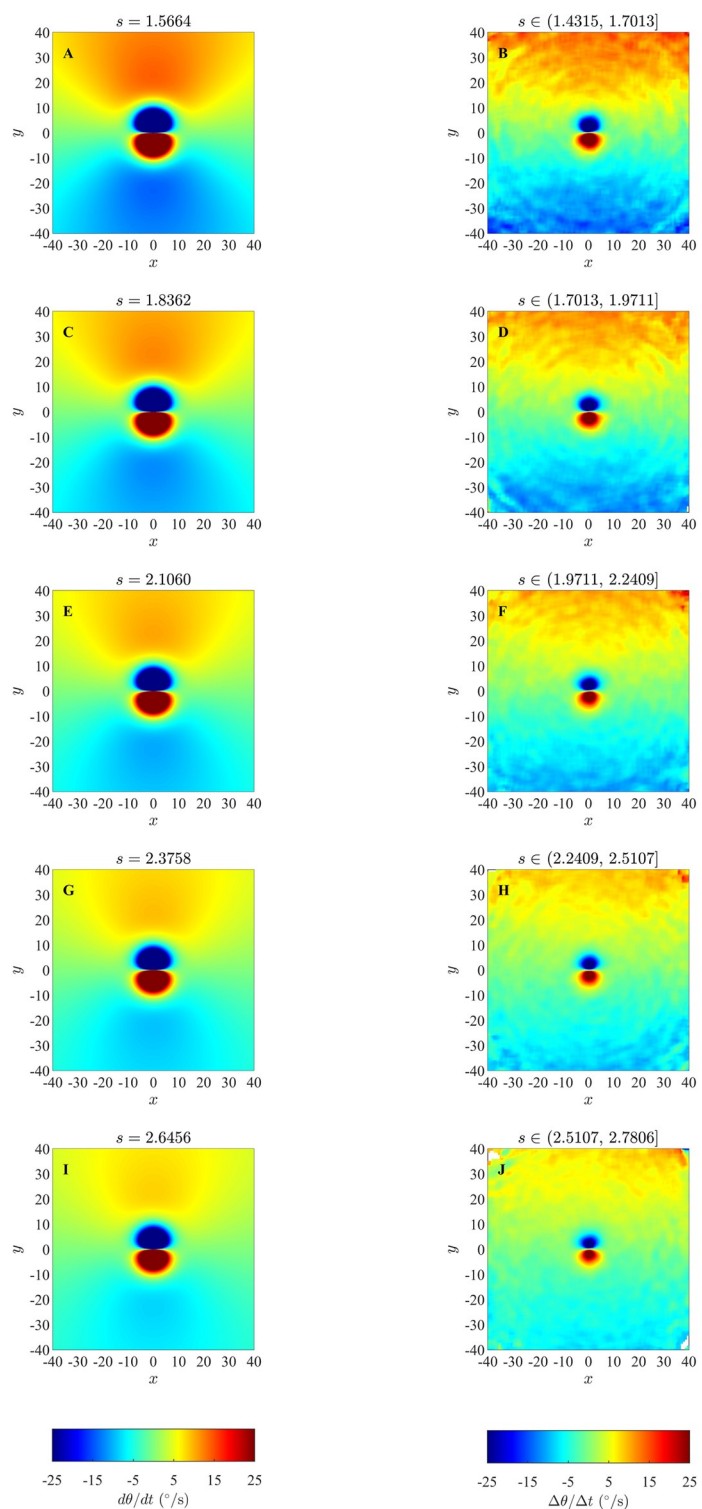

**Fig 8.** Left column: analytical pairwise turning interactions as prescribed by the ODE model (equation (S3.19)) for groups that form swarms (item (b) from Table 3). Right column: results obtained via the averaging method. Here the independent variables are relative partner positions (within each graph) and the speed of the focal individual (which varies across the five panels above). In each of the graphs the focal individual is located at the origin and moving right along the positive $x$-axis. Positive changes in angle/direction correspond to anti-clockwise/left turns, whereas negative changes in angle/direction correspond to clockwise/right turns. White regions indicate that no partner individuals were recorded with the corresponding relative $(x, y)$ coordinates for the given range of speed for focal individuals. (Derived from simulations with $N = 10$ individuals over 10000 time steps).

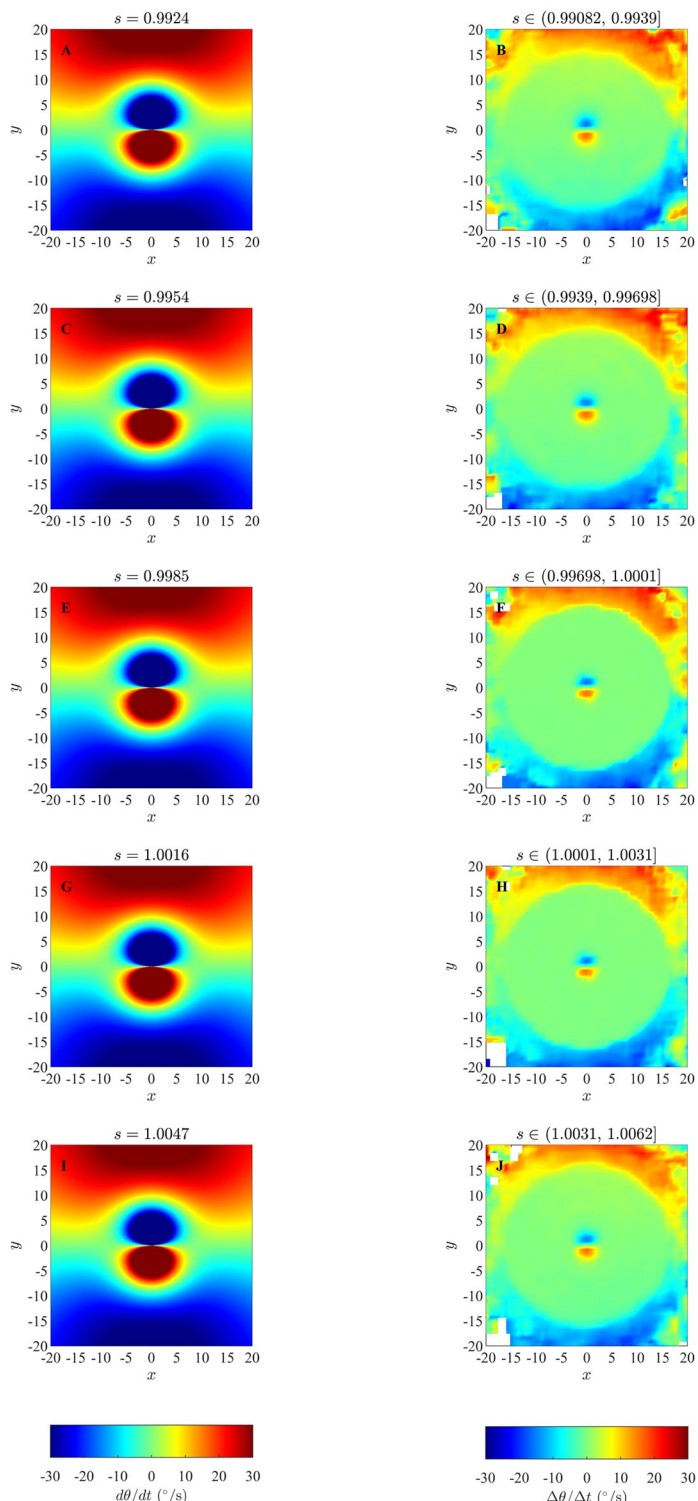

**Fig 9.** Left column: analytical pairwise turning interactions as prescribed by the ODE model (equation (S3.19)) for groups that move in parallel (item (c) from Table 3). Right column: results obtained via the averaging method. (Derived from simulations with $N = 10$ individuals over 10000 time steps).

## 4.4 Changes in speed in the ODE model

Results of our analysis for changes in speed of individuals as a function of relative positions of partners for the ODE model, as listed in Table 3 are shown in the Figs 10 and 11, and S37 to S41 Figs in S1 File. The averaging method consistently revealed the tendency for individuals to reduce their speed when their partners occupied the region immediately in front of them (blue regions immediately to the front of the focal individual in the right columns of Figs 10 and 11, and S37 to S41 Figs in S1 File) and increase their speed when partners occupied the region immediately to their rear (red regions immediately behind the focal individual). Hence the averaging method is capable of resolving the qualitative form of speed mediated collision avoidance at close range that is prescribed by the ODE model. Similarly, the averaging method also consistently revealed the tendency for individuals to increase their speed when partners were further to their front (redder regions from intermediate to greater distances to the front of the focal individual in the right columns of Figs 10 and 11, and S37 to S41 Figs in S1 File), and decrease their speed when partners were further to the rear (bluer regions to the rear of focal individuals). Thus the averaging method is capable of resolving the qualitative form of longer-range attraction to partners, moderated by changes in speed, at least for the cases studied here. However, analogous to the results for the analysis of changes in direction, the averaging method consistently identified a smaller region over which repulsion-like effects applied as compared to exact pairwise interactions (left columns of Figs 10 and 11, and S37 to S41 Figs in S1 File), and the size of the apparent repulsion region diminished as group size increased (S47 to S51 Figs in S1 File). In addition, the magnitudes of both increases and decreases in speed determined by the averaging method tended to be smaller than those expected from pairwise interactions.

## 5 Discussion and conclusion

Our graphs illustrating changes in direction of an individual as a function of the relative positions of groupmates were sufficiently accurate that they gave a reasonable indication of the radius of the prescribed zone of repulsion for a given set of simulated data. These graphs also captured the qualitative form of model-prescribed turning-based collision avoidance, illustrating that individuals turn away from near neighbours in a manner consistent with model interactions. In addition to providing a reasonable representation of the zone of repulsion, and the sense of turning responses to individuals in this region, when an individual's blind zone extended across the zones of repulsion, orientation, and attraction (with corresponding blind angle $\omega_{blind} = 90°$), then the blind zone was evident in the averaging method deduced zone of repulsion. This was most clearly seen when comparing analyses of simulations with or without blind zones; the region over which repulsion based turning behaviour manifested in our graphs was approximately circular when there was no blind zone (Figs 3B & 3C and 4B, S6B, S7B & S7C, S8B, S9B, S10B, S11B and S12B Figs in S1 File), whereas a sector of this circle immediately behind the focal individual was absent when individuals had a blind zone (Figs 3D & 3E and 4C, S6C Fig in S1 File (more apparent in S26 Fig in S1 File), S7D Fig in S1 File (more apparent in S27A Fig in S1 File), S8C & S8D Fig in S1 File (more apparent in S27B and S28A Figs in S1 File respectively), S9C, S10C, S11C & S11D, and S12C Figs in S1 File). There was sufficient detail in the wedge shaped region that simple graph based analysis of this region then led to a reasonable estimate of the angular extent of the blind zone. If the blind zone was only prescribed across individuals' zones of orientation and attraction then our analysis did not clearly capture the blind zone's form or presence. In related work, Heras et al. [39] found that an artificial neural network approach to fitting the functions that describe how an

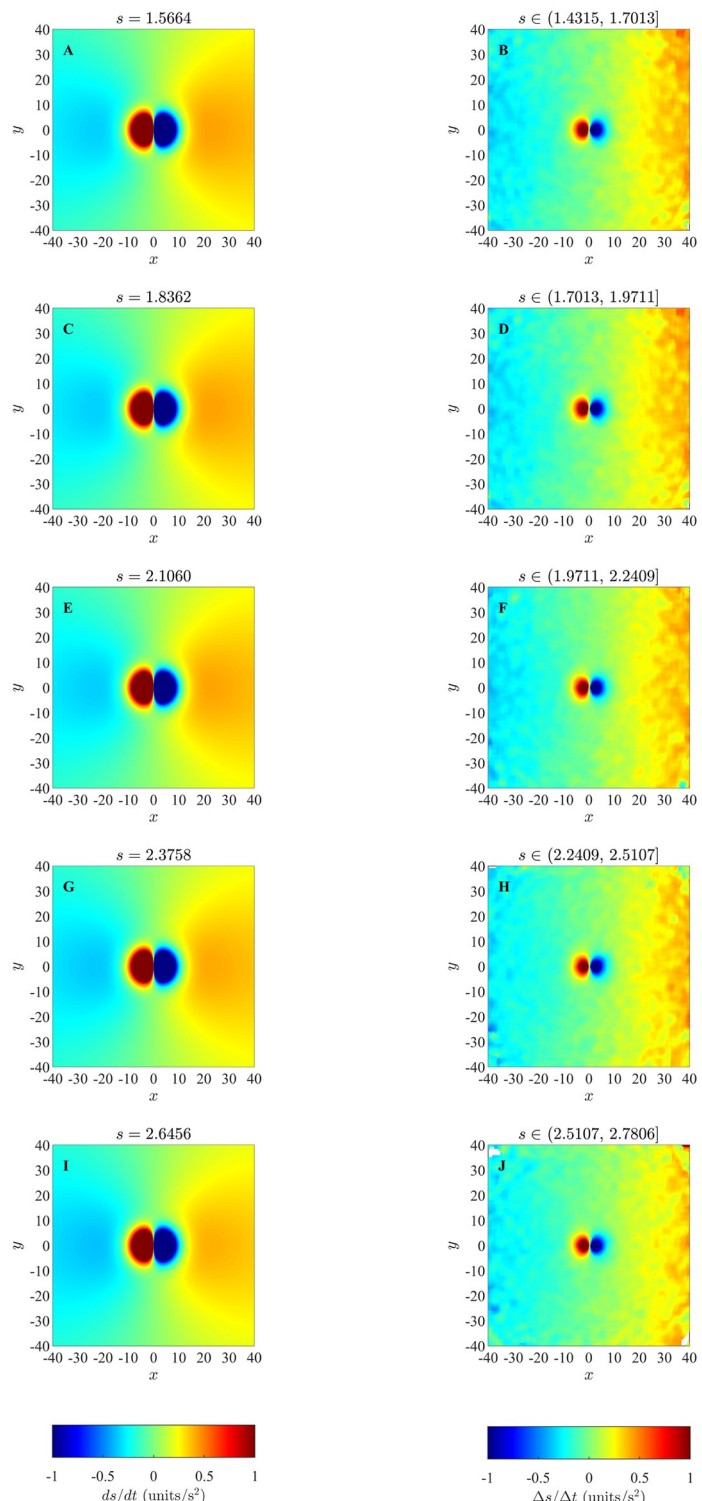

**Fig 10.** Left column: analytical pairwise changes in speed as prescribed by the ODE model (equation (S3.18)) for groups that form swarms (item (b) from Table 3). Right column: results obtained via the averaging method. Note that unlike changes in direction, changes in speed do not vary as a function of the speed of the focal individual, and thus there is no variation in the plots in the left column, and we expect little variation between the plots in the right column. In each of the plots, the focal individual is located at the origin and moving right along the positive $x$-axis. The focal individual increases its speed when its partners occupy redder regions in the graphs, and decreases its speed when its

partners occupy bluer regions. White regions indicate that no partner individuals were recorded with the corresponding relative $(x, y)$ coordinates for the given range of speed for focal individuals. ($N = 10$, 10000 time step simulations).

individual adjusts its velocity in response to the relative positions and behaviours of group mates could also reveal the details of a blind zone.

Although the analysis that we used did not explicitly target orientation interactions of the sort prescribed in the zonal model, where individuals were to turn to match directions of motion of their groupmates within a particular range of distances, there may have been indicators of the zone of orientation. In particular, when the emergent pattern of collective motion was parallel aligned, the zone of repulsion was surrounded by a larger almost annular region, predominantly green in colour (indicating average turning rates closer to zero for focal individuals), that was of similar radius to the radius of orientation (Fig 3C, 3E & 3G, S7C, S8B, S8D & S8F and S11B, S11D & S11F Figs in S1 File). However, subsequent inspection of our analysis of simulations of the ODE model revealed similar annular regions around the apparent repulsion zone, most prominently when groups moved in parallel, even though the ODE model does not prescribe any adjustments to the velocity of individuals based on the velocity of their partners. Hence the annular green regions in our analysis of the zonal model may in fact have been due to emergent group-level polarised patterns of movement, rather than indicators of explicit orientation zones.

If the emergent group level movement was one of cohesion (either milling or swarming), then an equivalent circular region was not present in our graphs. The absence of this circular region could have been because outside the repulsion zone, individuals adjust their motion based on an average of preferred directions of motion based on orientation and alignment. If the orientation zone has a relatively small width, as is the case for most swarms, then the attraction effect could tend to dominate, especially if the orientation zone is small enough to remain empty at relatively high frequency, and thus the sort of circular region that is correlated with the zone of orientation when group motion is polarised could be obscured (mills require a balance between orientation and attraction zone sizes, and might also fit this reasoning). A tendency to turn towards neighbours at some distance outside the zone of repulsion was evident in all our analyses of simulation data from the zonal model. However, the annular region over which pairwise attraction interactions applied was obscured, and attraction based turning tended to be detected across larger scales by the averaging method than those indicated by the prescribed radii of attraction zones.

In general, averaging based analysis of data from the ODE model consistently revealed the qualitative form of repulsion- and attraction-like behaviour of individuals in response to the relative locations of their partners (in both changes in direction and speed). However, the averaging method also consistently suggested that the size of the region over which repulsion-like interactions applied was smaller than in the exact case of pairwise interactions, and tended to suggest smaller magnitudes of changes in direction and speed than in the exact pairwise case. A possible reason for the reduction in the size of the zone of repulsion could be because the prescribed interactions of individuals, both repulsive and attractive, are added irrespective of the location of partners (Eq (3.1)), meaning that attraction-like behaviour towards individuals at greater distances can be added to, and cancel out, repulsion away from nearer neighbours. The effect could then be the apparently diminished size of the repulsion zone as compared to the exact form of pairwise interactions, due to interactions with multiple partners. S42 to S51 in the S1 File illustrate that the apparent size of the repulsion zone does in fact decrease consistently as group size increases. In contrast, the zonal model treats repulsion based interactions

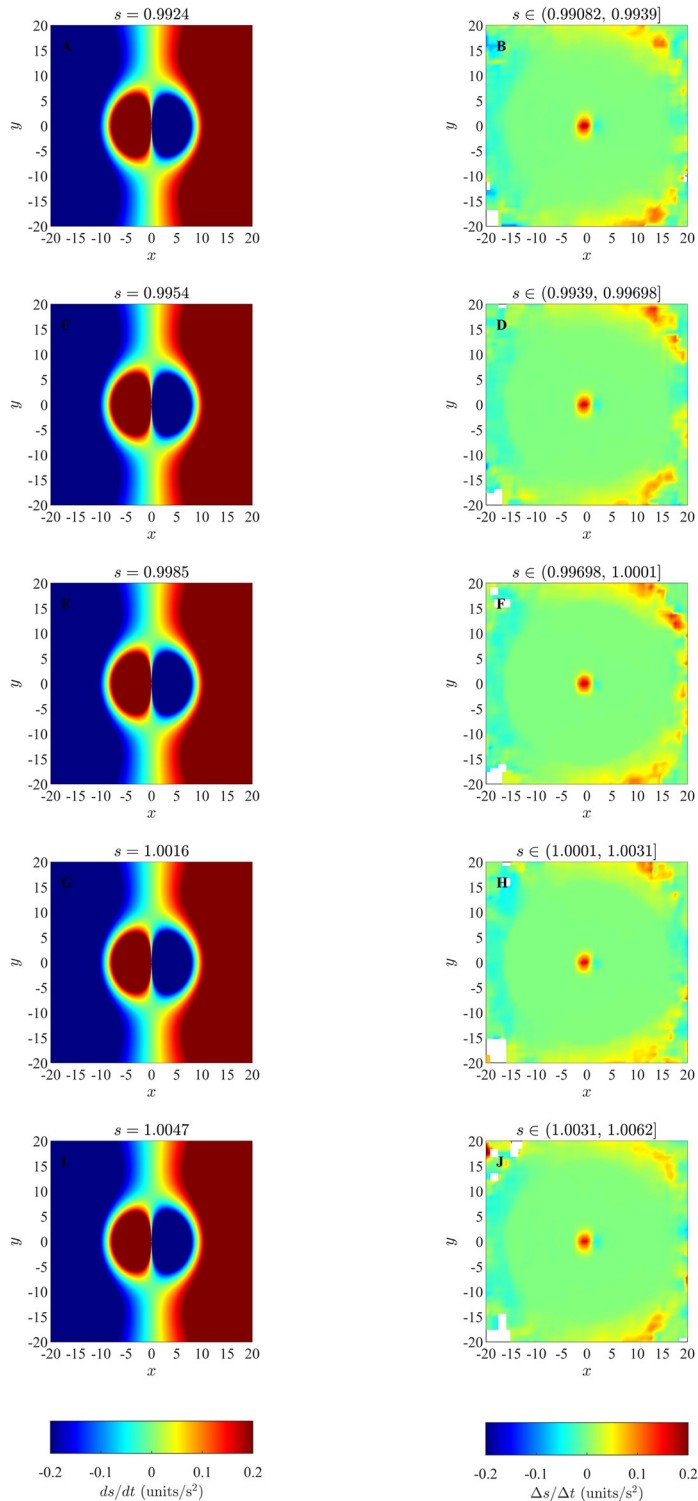

**Fig 11.** Left column: analytical pairwise changes in speed as prescribed by the ODE model (equation (S3.18)) for groups that undergo parallel motion (item (c) from Table 3). Right column: results obtained via the averaging method. (Derived from simulations with $N = 10$ individuals over 10000 time steps).

preferentially, and repulsion effects exclude any interactions with partners outside the repulsion zone, with the result seemingly that there is good correlation between the prescribed size of the zone of repulsion and that suggested by our averaging analysis. Fig 6 and S22 Fig in S1 File suggest little, or no, change in the size of the zone of repulsion as group size increases.

There seems to be limited literature that examines the effects of group size on interactions inferred by averaging methods, with the exceptions of two of the earlier studies that applied these methods [30, 31]. Katz et al. [30] examined interactions in groups of two and three golden shiners. One of the key findings of the work of [30] was that interactions in response to two other group members were not the average of pairwise interactions between focal individual and each partner in turn. Such a result suggests that the sort of aggregation of data across multiple individuals that we applied for this study might not perfectly reflect the nature of pairwise interactions, and that was in fact the case for our work.

Herbert-Read et al. [31] examined interactions in groups of two, four, and eight eastern mosquitofish, applying earlier variants of the methods used in our study, which included aggregation of data across multiple focal individuals. The qualitative nature of the interactions inferred did not vary as group size increased, with consistent evidence that mosquitofish moderated their speed to avoid collisions at short range, and to maintain contact with the group at greater distances ([31], Fig 2B and S5 Fig in S1 File (top row)). Individual mosquitofish also consistently turned towards neighbours ([31], Fig 2F and S5 Fig in S1 File (bottom row)), although more recent analysis at finer spatial resolution suggests a tendency for eastern mosquitofish to turn away from near neighbours to their front [46]. The qualitative nature of interactions with just first, second, or third nearest neighbours in groups of four in ([31], S4 Fig in S1 File) were not different to those determined by aggregation of data. However, there were some differences in detail in the interactions identified across groups of different sizes in [31], including diminution in the magnitudes of changes in speed and direction of motion as group size increased, and incremental increases in the size of the apparent zone of repulsion ([31], S5 Fig in S1 File).

Given that both the prior studies of [30, 31], and our study here indicate that group size has an effect on the quantitative elements of interactions inferred via an averaging method when the fitted function has only two independent variables, $x$ and $y$, it seems that it will be of value to investigate group size effects on such analysis in more detail in future studies. The approach used by Heras et al. [39] to infer interactions from groups of more than two included separation of elements of their analysis into modules. The modules included examinations of pairwise interactions, and functions that described the relative weights applied by a focal individual in combining pairwise interactions across multiple partners. An improved methodology for combining pairwise interactions, like that examined by Heras et al. [39], within the framework of an averaging method may also help to better deal with, and illuminate, what we think are the effects of group size.

Also apparent from this study is that group level patterns of movement can have an effect on interactions inferred via the averaging methods that we used. This is most evident in plots illustrating changes in direction as a function of the relative coordinates of group mates where there was a consistent sense of rotation by the group, as was the case with clockwise and anticlockwise rotating mills (see S33 and S34 Figs in S1 File). In an analysis of group level measures of order [41] took into account, and subtracted, the translation of the group's centre of mass, the rotation of the group about the centre, and the group's dilatation (a measure of a group's tendency to expand or contract in synchrony) before calculating group order parameters. Such corrections offer an immediate avenue to try to improve the averaging methods used here, and will be an element of our future research.

In addition, the analysis of parallel groups that emerged from simulations of the zonal model suggested that individuals were turning away from neighbours located in their blind zone, but beyond the radius of the zone of repulsion. We suspect this abberation is due to individuals applying repulsion based turns away from neighbours in their zone of repulsion, but having this turning-away behaviour also recorded as a response to trailing neighbours at greater distances. In terms of more accurately capturing the prescribed interactions of the zonal model, a potential method for correcting this type of error would be to only bin data associated with neighbours in the zone of repulsion, when there are neighbours in this region. However, such a correction requires prior knowledge of the mechanics of the model, and probably would not be appropriate for the analysis of real data where the goal is to infer interactions that are unknown, other than an assumption that individuals adjust their velocity as a function of the relative coordinates of their neighbours. An alternative approach could be to try to adapt some of the approach used by [38] to treat the observed turning response of each individual as a sum of repulsion, orientation, and attraction interactions to the averaging method framework.

Based on the analysis presented here, averaging methods are capable of correctly identifying the qualitative form of changes in individuals' components of velocity as a function of the relative coordinates of group mates in two dimensions, even with relatively limited data, as demonstrated by our analysis of data from short duration simulations. Depending on how individuals interact with multiple partners, averaging methods can also produce reasonable estimates of the size and shape of regions over which particular forms of interaction occur, such as the size of the prescribed zone of repulsion, even when data is aggregated based on the relative coordinates of multiple partners at the same time. Averaging methods can also be used to identify if individuals have a simple blind zone, or not. However, it appears that both group size, and emergent patterns of group movement, can have negative effects on the quantitative accuracy of an averaging method, and there is a need to improve averaging methods to take into account such factors.

## Supporting information

**S1 File. The supporting information for this work includes additional information on: The application of binning in the averaging method, details of the self-propelled particle zonal simulation model used throughout the work, analytical results relating to pairwise interactions for both models, algorithmic classification of emergent states for longer duration realisations of the simulation model, the calculations used to estimate blind zones, and supplementary results from the analysis of both models.**
(PDF)

**S2 File.**
(M)

**S3 File.**
(M)

**S4 File.**
(M)

**S5 File.**
(M)

**S6 File.**
(M)

## Acknowledgments

We thank Norman Gaywood for his support for this project through his management of the Turing computational system at the University of New England, which was vital for the completion of this work. We thank Geoff Bratt for his elegant method for constraining turning angles in our implementation of the self-propelled particle model, and Yihong Du, Thomas Kalinowski, Peter Loxley and Matthew Cooper for their suggestions for an analytic point of reference for our calculations. We also thank the reviewers for their comments that helped to expand the scope of, and improve the presentation of, this work.

## Author Contributions

**Conceptualization:** Rajnesh K. Mudaliar, Timothy M. Schaerf.

**Formal analysis:** Rajnesh K. Mudaliar, Timothy M. Schaerf.

**Investigation:** Rajnesh K. Mudaliar, Timothy M. Schaerf.

**Methodology:** Rajnesh K. Mudaliar, Timothy M. Schaerf.

**Project administration:** Timothy M. Schaerf.

**Supervision:** Timothy M. Schaerf.

**Writing – original draft:** Rajnesh K. Mudaliar.

**Writing – review & editing:** Rajnesh K. Mudaliar, Timothy M. Schaerf.

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
