## [Decision Letter · Decision Letter 0]

26 Aug 2020

PONE-D-20-22104

Examination of an averaging method for estimating repulsion and attraction interactions in moving groups

PLOS ONE

Dear Dr. Schaerf,

Thank you for submitting your manuscript to PLOS ONE. After careful consideration, we feel that it has merit but does not fully meet PLOS ONE’s publication criteria as it currently stands. Reviewers are concerned with the format of the manuscript and suggest performing additional simulations. Therefore, we invite you to submit a revised version of the manuscript that addresses the points raised during the review process. We will then seek an additional round of review from at least one of the reviewers.

We look forward to receiving your revised manuscript.

Kind regards,

Ivan Kryven

Academic Editor

PLOS ONE

Journal Requirements:

"This work was supported in part by funding from the Australian Research Council's Discovery Project scheme, under project code DP190100660."

Reviewers' comments:

Reviewer's Responses to Questions

**Comments to the Author**

1. Is the manuscript technically sound, and do the data support the conclusions?

Reviewer #1: Yes

Reviewer #2: Yes

2. Has the statistical analysis been performed appropriately and rigorously? 

Reviewer #1: Yes

Reviewer #2: Yes

3. Have the authors made all data underlying the findings in their manuscript fully available?

Reviewer #1: Yes

Reviewer #2: Yes

4. Is the manuscript presented in an intelligible fashion and written in standard English?

Reviewer #1: Yes

Reviewer #2: Yes

5. Review Comments to the Author

Reviewer #1: Please see attached document for my review.

Please see attached document for my review.

Reviewer #2: The current article investigates the resolution and accuracy of force maps as a tool to understand collective behaviour. To properly benchmark this common tool, the authors have used 2 theoretical models of collective behaviour, and tried to extract some of the properties of the model, such as regions of repulsion, attraction and alignment.

While I have some harsh criticism in the form the authors chose to present this manuscript, they did present a valid and meaningful scientific study, even if it requires major revisions.

The manuscript was written in a "text book" or "thesis" format, where the authors defined and explained every single equation and procedure used in their paper, most of them multiple times! Even without a page limit, and knowing that papers should be written to a naive audience, the authors took this to the extreme. As a review, thesis or a class project, this paper is great, but as a publication, a very small proportion of people will be willing to read 8 pages (mostly of repetitive definitions) before even properly describing the models they used. The authors even describe their definition of the data binning. In short, the first  18 pages of the paper would normally have been summarised in 3 or 4 pages at most. Similarly, the number of figures in this paper feels excessive, normally most of them would be presented as SI material.

My second criticism of the paper comes in the quantity and duration of simulations the authors have used. 1000 time steps and 10 simulations for a zone model with 25 agents? I understand the authors have explored a large region of the parameter space, and that their results might not change significantly if they expand their stats, but with 1000 time steps one cannot even be sure the results are not a pure result of the initial conditions, and with only 10 simulations this is just unacceptable. I would suggest a minimum of 10^5 time steps, where the first half is discarded, and 100 or more simulations per condition. Similarly for the ODE model, increasing the length of each simulation by an order of magnitude, while noting that the number of simulations is never described in the text (but only in table 3).

A third criticism (but this is optional), was not to explore the effect of population size in the results, it felt like a missed opportunity.

One last possible issue comes from a publication [1] that came probably after the authors had submitted this paper. Considering the necessity of major revisions, I feel that the authors should also compare their work to this new one, in addition to Heras et al.

[1] Escobedo R., Lecheval V., Papaspyros V., Bonnet F., Mondada F., Sire C. and Theraulaz G. 2020A data-driven method for reconstructing and modelling social interactions in moving animal groupsPhil. Trans. R. Soc. B37520190380http://doi.org/10.1098/rstb.2019.0380

Smaller issues:

Figure 2 makes it clear that the radius of the regions are much larger than the maximum turning rate, meaning most of the interactions are given by the max value of the model. How do these values compare to the original Couzin publication? If they are similar, this could be a criticism of the model. Also, figure 2 has no units in the color label.

Lastly, the colors of the heatmaps are in general not colorblind friendly. While not essential, the final range of colors was just distracting, Figure 3 being the most extreme example.

6. PLOS authors have the option to publish the peer review history of their article (what does this mean?). If published, this will include your full peer review and any attached files.

Reviewer #1: No

Reviewer #2: No

---

## [Author Response · Author response to Decision Letter 0]

19 Nov 2020

I have attached our responses to the reviewers comments with the new letter to the editor.

---

## [Editor Report · Decision Letter 1]

25 Nov 2020

Examination of an averaging method for estimating repulsion and attraction interactions in moving groups

PONE-D-20-22104R1

Dear Dr. Schaerf,

We’re pleased to inform you that your manuscript has been judged scientifically suitable for publication and will be formally accepted for publication once it meets all outstanding technical requirements.

Kind regards,

Ivan Kryven

Academic Editor

PLOS ONE
---

## [Editor Report · Acceptance letter]

27 Nov 2020

PONE-D-20-22104R1

Examination of an averaging method for esti- mating repulsion and attraction interactions in moving groups

Dear Dr. Schaerf:

I'm pleased to inform you that your manuscript has been deemed suitable for publication in PLOS ONE. Congratulations! Your manuscript is now with our production department.

Kind regards,

on behalf of

Dr. Ivan Kryven

Academic Editor

PLOS ONE